# Structural Dynamics, Phonon Spectra and Thermal Transport in the Silicon Clathrates

**DOI:** 10.3390/molecules27196431

**Published:** 2022-09-29

**Authors:** Benxiang Wei, Joseph M. Flitcroft, Jonathan M. Skelton

**Affiliations:** Department of Chemistry, University of Manchester, Oxford Road, Manchester M13 9PL, UK

**Keywords:** silicon clathrates, structural dynamics, phonon spectra, infrared spectra, raman spectra, thermal conductivity, thermoelectrics, density-functional theory

## Abstract

The potential of thermoelectric power to reduce energy waste and mitigate climate change has led to renewed interest in “phonon-glass electron-crystal” materials, of which the inorganic clathrates are an archetypal example. In this work we present a detailed first-principles modelling study of the structural dynamics and thermal transport in bulk diamond Si and five framework structures, including the reported Si Clathrate I and II structures and the recently-synthesised *o*C24 phase, with a view to understanding the relationship between the structure, lattice dynamics, energetic stability and thermal transport. We predict the IR and Raman spectra, including ab initio linewidths, and identify spectral signatures that could be used to confirm the presence of the different phases in material samples. Comparison of the energetics, including the contribution of the phonons to the finite-temperature Helmholtz free energy, shows that the framework structures are metastable, with the energy differences to bulk Si dominated by differences in the lattice energy. Thermal-conductivity calculations within the single-mode relaxation-time approximation show that the framework structures have significantly lower $\kappa_{\mathrm{latt}}$ than bulk Si, which we attribute quantitatively to differences in the phonon group velocities and lifetimes. The lifetimes vary considerably between systems, which can be largely accounted for by differences in the three-phonon interaction strengths. Notably, we predict a very low $\kappa_{\mathrm{latt}}$ for the Clathrate-II structure, in line with previous experiments but contrary to other recent modelling studies, which motivates further exploration of this system.

## 1. Introduction

Among the most important challenges to contemporary science is meeting the ever-increasing worldwide demand for energy whilst reducing anthropogenic greenhouse gas emissions. Achieving this goal will require a suite of technologies including new and improved sources of clean energy together with improvements to the efficiency of energy-intensive processes.

An estimated 60% of the energy used globally is currently wasted as heat, which is only estimated to improve to around 50% by 2030, and this has led to intensive research interest in thermoelectric power [1,2]. Thermoelectric generators (TEGs) harness the Seebeck effect in a thermoelectric material to directly convert waste heat into electrical power [3]. TEGs are solid-state devices with no moving parts, and as such have a broad range of potential applications from powering remote sensing devices, to extracting waste heat from combustion engine exhaust trains, to repurposing decommissioned oil rigs as geothermal power plants [3,4].

The performance of a thermoelectric material is determined by the dimensionless figure of merit, ZT: [1].
(1)$$ZT=\frac{S^{2}\sigma T}{\kappa_{\mathrm{elec}}+\kappa_{\mathrm{latt}}}$$
where *S* is the Seebeck coefficient, σ is the electrical conductivity and κ is the thermal conductivity, which is the sum of the electronic and lattice (phonon) components $\kappa_{\mathrm{elec}}$ and $\kappa_{\mathrm{latt}}$. *S*, σ and $\kappa_{\mathrm{elec}}$ all depend on the electronic structure of the material and are linked through the carrier concentration, *n*, such that the best balance is typically found in heavily-doped semiconductors [1]. In particular, increasing *n* increases the conductivity but generally decreases the Seebeck coefficient and increases the electronic contribution to the thermal conductivity, which in heavily-doped semiconductors is often proportional to σ through the Wiedemann–Franz law. The $\kappa_{\mathrm{latt}}$ depends on the structure and chemical bonding of the material, and low $\kappa_{\mathrm{latt}}$ is typically found in materials composed of heavy elements with weak chemical bonding [5] and/or strongly anharmonic lattice dynamics associated with structural features such as phase transitions [6,7,8] and active lone pairs [9].

The requirement to optimise both the electronic transport and the lattice thermal conductivity has historically led to a focus on “heavy” chalcogenide materials, which include the current industry standards for room- and high-temperature applications Bi_2_Te_3_ (ZT ≃ 1 between 350–450 K) and PbTe (ZT ≃ 2.2 at 915 K with endotaxial nanostructuring) [10]. However, an issue with these types of materials is the toxicity of heavy metals such as Pb and the rarity of Te, both of which mean these current “champion” materials are unsuitable for mass production and are therefore restricted to niche applications. This has led to wide-ranging research to identify more abundant and less toxic candidate thermoelectrics, including other chalcogenides, oxides and oxyclalcogenides alongside skutterudite and clathrate framework structures, intermetallics and half-Heuslers, among others [3,11,12,13].

Approaches to improving ZT fall into two general classes, viz. optimising the power factor $S^{2}\sigma$ by tuning the electronic structure e.g., using band engineering [1,14,15], and reducing the lattice thermal conductivity e.g., through alloying, “discordant atom” doping, and hierarchical nanostructuring [10,16,17,18,19]. The interlinked nature of the transport properties makes optimisation challenging: for example, a high electrical conductivity requires a low carrier effective mass m∗ (high carrier mobility μ), whereas a large Seebeck coefficient is favoured by a large m∗, and the conductivity and electronic thermal conductivity are typically proportional such that increasing σ also increases $\kappa_{\mathrm{elec}}$ [1]. On the other hand, the lattice thermal conductivity is largely independent and can in principle be minimised without degrading the electrical transport [1]. However, the comparatively limited microscopic understanding of heat transport through phonons and how it can be controlled means that at present this is largely done through empirical strategies and trial and error.

The dual requirement for good electrical transport characteristic of crystalline semiconductors and poor thermal transport characteristic of amorphous materials (glasses) is encapsulated in the “phonon glass electron crystal” (PGEC) concept initially proposed by Slack [20]. A well-known subclass of PGEC compounds are materials based on a crystalline framework into which loosely-bound guest species can be incorporated. The “rattling” of the guest species in the cages couples to the framework modes and suppress the thermal transport through the lattice [21], while the electronic transport through the framework is unaffected, and the guest species can also be used to optimise the electrical properties e.g., by reducing or oxidising framework atoms to control the carrier concentrations. This idea has been used to good effect in CoSb_3_, where the ZT of 0.05 at 723 K shown by the pristine material [22] can be increased by filling with single ions (e.g., Na_0.48_Co_4_Sb_12_ − ZT = 1.32 at 800 K [23]) or multiple species (e.g., Ba_0.08_La_0.05_Yb_0.04_Co_4_Sb_12_ − ZT = 1.7 at 850 K [24]).

A second subclass of potential PGEC compounds are the inorganic clathrates typically formed by Group 14 elements. Clathrates are “open-structured” 3D frameworks composed of polyhedral cages, and the wide variety of possible structural motifs lead to a complex relationship between the cage structure, structural dynamics, and the thermal conductivity. Clathrates show potential for thermometric applications because of their transport properties, wide band gap [25,26,27] and reported low thermal conductivity [25,26,27]. Elemental Si adopts a bulk diamond structure, which can be easily doped to obtain a competitive power factor of around 50 μW K${}^{-2}$ between ∼800–1200 K [28,29] but has a high $\kappa_{\mathrm{latt}}$ on the order of ∼25–45 W m${}^{-1}$ K${}^{-1}$ over the same temperature range [30,31]. However, Si can also form a number of clathrate phases [32,33,34] and other open-framework structures [35,36], at least one of which has been measuered to have a much lower thermal conductivity than diamond Si [37]. Ab inito calculations are playing an increasingly important role in building our microscopic understanding of lattice thermal conductivity, providing insight into both the origin of the intrinsically-low $\kappa_{\mathrm{latt}}$ of high-performance thermoelectrics such as SnSe [7,19,38] and also how material modifications such as alloying and filling can be used to optimise it [18,39,40]. Given the potential of clathrate structures in general, it is of considerable interest to use these techniques to unravel the link between the structure and thermal conductivity in the allotropes of Si.

In this work we utilise periodic density-functional theory (DFT) calculations to investigate the structural dynamics of bulk diamond Si and five framework structures drawn from experiments and previous theoretical studies [32,35,36]. We model and analyse the phonon spectra and compute reference infrared (IR) and Raman spectra for confirming the identity of the materials in experimental syntheses. We compare the energetic stabilities using both the athermal lattice energies and the constant-volume Helmholtz free energies including the phonon contributions to the internal energy and entropy. Finally, we calculate and analyse the lattice thermal conductivities of the six materials and elucidate the microscopic origin of the significant differences in the heat transport of the bulk and framework structures. From these calculations we obtain a detailed understanding of how moving from a dense bulk phase to open framework structures affects the structural dynamics and thermal transport, and we hope the improved fundamental understanding of the factors that determine the thermal transport in clathrates and related systems will inform the future design of high-performance thermoelectrics with ultra-low $\kappa_{\mathrm{latt}}$.

## 2. Computational Modelling

Periodic density-functional theory calculations were performed using the Vienna Ab initio Simulation Package (VASP) code [41].

As described in Section 3.1, initial structures of bulk diamond Si and five framework structures were obtained from the Materials Project database [42] and the Inorganic Crystal Structure Database (ICSD).

The PBEsol functional [43] was used to describe electron exchange and correlation, as we have previously found that this gives good results for lattice dynamics and thermal conductivity [44]. The Si atoms were described using projector augmented-wave (PAW) pseudopotentials [45,46] treating the Si 3s and 3p electrons as valence states. The Kohn-Sham wavefunctions of the valence electrons were represented in a plane-wave basis with a kinetic-energy cutoff of 500 eV. Brillouin-zone integrations were performed using the Γ-centered Monkhorst-Pack k-point meshes [47] listed in Table 1 and a Gaussian smearing with a width σ of 0.01 eV. The cutoff and k-point meshes were both chosen to converge the absolute total energy and cell pressure to <1 meV atom${}^{-1}$ and <1 kbar (0.1 GPa) respectively. The size of the charge-density grids was determined automatically to avoid aliasing errors (PREC = Accurate in VASP), and non-spherical contributions to the gradient corrections inside the PAW spheres were accounted for (LASPH = .TRUE.).

Each structure was symmetrised using spglib [48] and fully optimised to tolerances of 10${}^{-8}$ eV on the electronic total energy and 10${}^{-2}$ eV Å${}^{-1}$ on the forces.

Lattice-dynamics and thermal-conductivity calculations were set up and post-processed using the Phonopy and Phono3py software [49,50]. The second- and third-order force constants were determined using supercell finite-displacement calculations with the supercells shown in Table 1 and the default displacement step sizes of 10${}^{-2}$ and 3 × 10${}^{-2}$ Å respectively. Where required conversions between primitive and conventional cells were performed using spglib and/or by applying appropriate appropriate transformation matrices during post-processing. During the single-point force calculations additional support grids with 8× the number of points as the standard charge-density grids were employed to reduce numerical noise (ADDGRID = .TRUE.).

Phonon density of states (DoS) curves g(ν) were computed by interpolating the phonon frequencies onto the regular Γ-centered q-point grids listed in Table 1. The DoS curves were broadened using a Gaussian smearing with a width σ = 0.064 THz, corresponding to a full-width at half-maximim (FWHM) of 5 cm${}^{-1}$. These meshes were also used to compute the phonon contributions to the constant-volume (Helmholtz) free energy $A_{\mathrm{vib}}$. Phonon dispersion curves ν(q) were computed by interpolating frequencies along strings of q passing between high-symmetry wavevectors in the respective Brillouin zones. The lattice thermal conductivities were computed using the single-mode relaxation-time approximation (SM-RTA) [50] from modal properties evaluated on the regular q-point grids listed in Table 1.

Finally, the infrared (IR) and Raman activities of the Γ-point phonon modes were computed using the procedure outlined in Refs [51,52]. The Born effective-charge tensors $Z^{*}$ required for the IR activities were calculated using the density-functional perturbation theory (DFPT) routines in VASP [53]. The polarisability derivatives $\partial \alpha /\partial Q\equiv \partial \varepsilon^{\infty }/\partial Q$ were computed using numerical differentiation with a step size ΔQ chosen such that the largest Cartesian displacement was 10${}^{-2}$ Å. The $\varepsilon^{\infty }$ were computed using DFPT, with the exception of two modes of the C-II structure for which the DFPT calculation failed to converge and we therefore used instead the finite-field method outlined in Ref. [54] with a field strength of 10${}^{-3}$ eV Å${}^{-1}$.

## 3. Results and Discussion

### 3.1. Selection of Framework Structures

A search of the Materials Project (MP) database [42] yielded 39 entries for Si. 25 entries have Inorganic Crystal Structure Database (ICSD) codes and associated publications, of which 13 are labelled as being experimentally-determined and 12 as theoretical structures. The remaining 14 structures are user submissions with little other identifying information.

From the ICSD structures marked as experimental we selected the ambient diamond structure of bulk Si (**mp-149**), the Type-II clathrate structure published with Ref. [32] (**mp-16220**) and the “*o*C24” framework structure reported in Ref. [36] (**mp-1095269**). The *o*C24 structure is not formally designated as a clathrate but has a similar cage-like structure and is formed in a similar manner to the Type-I and Type-II clathrates [33,36,55]. The other structures include the hexagonal diamond structure that can be prepared in nanowire form [56,57] and a number of dense phases formed under pressure [58,59,60,61,62], neither of which are of interest to the present study.

The theoretical study in Ref. [35] proposes 11 so-called “Kelvin” framework structures. From these, we selected the three lowest-energy structures, viz. the ${\mathrm{Si}}_{K}^{\mathrm{VII}}$ structure reported as the Type-V clathrate in Ref. [34] (**mp-1203790**), the ${\mathrm{Si}}_{K}^{\mathrm{II}}$ structure identified as corresponding to the empty Type-I clathrate (**mp-971662**), and the ${\mathrm{Si}}_{K}^{V}$ structure similar to the Type-VI structure reported in Ref. [34]. The MP entry for the latter (**mp-1196961**) has a different spacegroup and cell volume to the linked ICSD structure, so we instead downloaded this structure from the ICSD (**ICSD: 189396**).

Table 2 lists the six structures selected for calculations together with their spacegroups, optimised lattice parameters, and the labels used to identify them. Corresponding images of the optimised structures are shown in Figure 1. The optimised lattice parmeters are generally in good agreement with experimental measurements and other theoretical literature. The room-temperature lattice constant of bulk Si is 5.431 Å [63], to which our calculated value of 5.436 Å is a good match. Our predicted lattice constants for the *o*C24 structure are *a* = 3.826, *b* = 10.70 and *c* = 12.66, which similarly compare well to the measured values of *a* = 3.82, *b* = 10.70 and *c* = 12.63 Å [36]. For the C-II structure we predict a lattice constant of 14.63 Å, which is very close to the measured 14.62 Å reported in the original structural characterisation in Ref. [64]. Our predicted volume of 23.02 Å${}^{3}$ atom${}^{-1}$ also lies between the values of 22.65 and 23.49 Å${}^{3}$ atom${}^{-1}$ obtained from the calculations in Ref. [32]. Likewise, our predicted volumes for the K-II/C-I, K-V/C-VI and K-VII/C-V structures are within 2% of the volumes calculated in Ref. [35].

### 3.2. Phonon Spectra

The calculated phonon dispersion ν(q) and density of states (DoS) curves g(ν) of the six structures are compared in Figure 2. The primitive cell of bulk diamond Si has a simple unit cell containing $n_{a}$ = 2 atoms, resulting in 3$n_{a}$ = 6 branches at each wavevector q in the dispersion, and the high symmetry of the $Fd\bar{3}m$ spacegroup is such that two pairs of the branches are degenerate along the path shown in Figure 2a. In comparison, the more complex structures of the five framework materials result in larger primitive cells and thus more complex phonon dispersion curves, with 3$n_{a}$ = 102 (C-II), 36 (*o*C24), 138 (K-II/C-I), 120 (K-V/C-VI) and 204 branches (K-VII/C-V).

The three acoustic modes in d-Si show a wide dispersion, reaching up to around 400 cm${}^{-1}$ at the zone-boundary wavevectors, while the optic modes span a relatively narrower range of ∼150 cm${}^{-1}$. In the five framework structures, the acoustic modes overlap with low-lying optic modes towards the zone boundaries, making it difficult to determine the frequency range precisely, but the acoustic modes do not appear to reach frequencies above ∼200 cm${}^{-1}$ and therefore have a considerably narrower dispersion than in d-Si. Each of the framework structures shows groups of optic modes both with relatively flat and with wider dispersions that are in some cases comparable to the optic modes in d-Si.

Despite the variations in complexity, the phonon frequencies in all six structures span a similar range of ∼500 cm${}^{-1}$. To assess the differences in the distribution of phonon frequencies over this range, we compared the cumulative integral of the phonon density of states as a function of frequency (Figure 3). Interestingly, we find that the phonon spectrum of the C-II structure shows a marked shift to lower frequencies compared to the other five systems, which otherwise have similar frequency distributions. This difference indicates that the C-II structure has significantly weaker chemical bonding compared to the other structures, a point to which we return in the following subsections.

While the dispersion and DoS can be measured using inelastic X-ray or neutron scattering (IXS/INS), these are relatively complex experiments that usually require access to synchrotron X-ray or neutron sources. Since the wavelengths of modes close to the Brillouin zone centre q = Γ are comparable to those of visible and infrared (IR) light, these modes can also be probed using more routine IR and Raman spectroscopy. Simulated spectra can be generated from these calculations as a weighted sum of Lorentzian peak functions according to: [52]
(2)$$I\left(\nu \right)=\sum_{q=\Gamma ,j}\frac{I_{j}}{\pi }\frac{\frac{1}{2}\Gamma_{j}}{{\left(\nu -\nu_{j}\right)}^{2}+{\left(\frac{1}{2}\Gamma_{j}\right)}^{2}}$$
where the sum runs over the 3$n_{a}$ modes *j* at q = Γ with ordinal frequencies $\nu_{j}$, spectroscopic activities $I_{j}$ and linewidths $\Gamma_{j}$, and we have dropped the q subscript from the terms in the summand for brevity.

The IR activity depends on the change in polarisation as the atoms are displaced along the mode. Given a Γ-point mode eigenvector $W_{j}$ the Cartesian displacement $X_{j,k}$ of the *k*th atom is given by:(3)$$X_{j,k}=\frac{1}{\sqrt{m_{k}}}W_{j,k}$$
where $m_{k}$ is the atomic mass. The infrared activities $I^{\mathrm{IR}}$ are given by the change in polarisation along the mode eigenvector and are calculated from the Born effective-charge tensors of the atoms $Z_{k}^{*}$ using the expression: [51]
(4)$$I_{j}^{\mathrm{IR}}=\sum_{\alpha }{\left|\sum_{k}\sum_{\beta }Z_{k,\alpha \beta }X_{j,k,\beta }\right|}^{2}$$
where α and β are the Cartesian directions *x*, *y* and *z*.

The Raman activity $I^{\mathrm{Raman}}$ depends on the change in polarisability α, which can be computed from the derivative of the high-frequency dielectric constant $\varepsilon^{\infty }$ with respect to the mode amplitude *Q*: [51]   
(5)$$I_{j}^{\mathrm{Raman}}=\frac{V}{4\pi }\frac{\partial \alpha }{\partial Q_{j}}\equiv \frac{V}{4\pi }\frac{\partial \varepsilon^{\infty }}{\partial Q_{j}}$$
where *V* is the unit-cell volume. The Raman activities are tensor quantities and the $I_{j}$ therefore depend on the measurement geometry. To account for this, we perform an orientational average to obtain the scalar intensities ${\bar{I}}^{\mathrm{Raman}}$ that would be measured from a powder sample: [51]
(6)$$\begin{array}{cc} {\bar{I}}^{\mathrm{Raman}}= & 45{\left[\frac{1}{3}\left(I_{11}+I_{22}+I_{33}\right)\right]}^{2}+\\ \; & \frac{7}{2}\left[{\left(I_{11}-I_{22}\right)}^{2}+{\left(I_{11}-I_{33}\right)}^{2}+{\left(I_{22}-I_{33}\right)}^{2}+6\left(I_{12}^{2}+I_{13}^{2}+I_{23}^{2}\right)\right] \end{array}$$
where $I_{\alpha \beta }$ denote the components of the Raman tensor.

In this work we compute the linewidths $\Gamma_{j}$ from the imaginary part of the phonon self-energy from many-body perturbation theory using the method documented in Ref. [50]. These are the intrinsic linewidths from energy- and momentum-conserving three-phonon scattering processes and do not include any instrument broadening.

The simulated IR and Raman spectra of the six materials are compared in Figure 4. The irreducible representations $\Gamma_{3N}$ spanned by the Γ-point modes in the six systems, based on which we identified the expected number of IR- and Raman-active modes, are summarised in Table 3.

The cubic $Fm\bar{3}m$ and $Pm\bar{3}m$ spacegroups adopted by d-Si, C-II and K-II/C-I both have the $O_{h}$ point group in which the T1u modes are IR active and the A1g, Eg and T2g modes are Raman active. In all three systems, the three acoustic modes form a T1u set. Although potentially IR active, since these modes correspond to rigid translations that cannot induce a change in polarisation they are constrained to have zero intensity. d-Si therefore shows no IR activity and a single peak in the Raman spectrum. We compute a central frequency of 508 cm${}^{-1}$ and a room-temperature (300 K) linewidth of 3.3 cm${}^{-1}$ for the Raman peak, which compare well to the experimental measurements of 520 cm${}^{-1}$ and 4.6 cm${}^{-1}$ respectively. [66] The much larger numbers of modes of the C-II and K-II/C-I structures result in more featured spectra with distinctive IR and Raman bands. The peak linewidths in the C-II spectrum are notably broader than in the other five spectra, which is indicative of stronger phonon anharmonicity and shorter lifetimes. We return to this point in Section 3.4.

The orthorhombic Cmcm and Cmmm spacegroups adopted by the *o*C24 and K-V/C-VI structures have the $D_{2h}$ point group where the B1u, B2u and B3u modes are IR active and the Ag, B1g, B2g and B3g modes are Raman active. The acoustic modes span a representation of B1u + B2u + B3u, so again a subset of the IR-active modes are restricted to have zero intensity. The *o*C24 structure has a considerably smaller unit cell than K-V/C-VI and as such has fewer modes, resulting in a notably less featured IR spectrum. On the other hand, although the K-V/C-VI structure potentially has ∼3× the number of Raman-active modes as the *o*C24 phase, differences in the relative intensities mean the spectrum of the latter shows more prominent features.

The $P6_{3}/mmm$ spacegroup of the K-VII/C-V structure has the $D_{6h}$ point group in which the A2u and E1u modes are IR active, the A1g, E1g and E2g modes are Raman active, and the acoustic modes span a representation of A2u + E1u. Of the six structures examined in this work K-III/C-V has the largest unit cell and the largest number of modes, although symmetry restrictions mean that there are fewer IR and Raman active modes than in the orthorhombic K-V/C-VI structure. The net result is that the spectra of these two structures are of comparable complexity. The Raman spectrum again has fewer prominent features than the *o*C24 structure due to differences in the relative intensities.

Overall, our simulations suggest that the differences in the six spectra would be sufficient to identify the different phases experimentally, especially using data from both IR and Raman spectroscopy. To aid with experimental characterisation and spectrum assignment, we provide a set of peak tables listing the central frequencies, irreducible representations, IR/Raman activities and linewidths calculated for the six structures as Appendix A.

### 3.3. Thermodynamics

Within the harmonic approximation the temperature-dependent constant-volume (Helmholtz) free energy A(T) is given by:(7)$$A\left(T\right)=U_{\mathrm{latt}}+A_{\mathrm{vib}}\left(T\right)=U_{\mathrm{latt}}+U_{\mathrm{vib}}\left(T\right)-TS_{\mathrm{vib}}\left(T\right)$$
where $U_{\mathrm{latt}}$ is the lattice internal energy, equal to the DFT total energy obtained after structural optimisation, and $A_{\mathrm{vib}}=U_{\mathrm{vib}}-TS_{\mathrm{vib}}$ is the vibrational free energy from contributions of the phonons to the free energy and entropy.

$A_{\mathrm{vib}}$ can be computed directly from the vibrational partition function $Z_{\mathrm{vib}}$ using the bridge relation:(8)$$A_{\mathrm{vib}}\left(T\right)=-\frac{1}{N}k_{B}T\ln\left[Z_{\mathrm{vib}}\left(T\right)\right]$$
where *N* is the number of wavevectors q included in the partition function product-equivalent to the number of unit cells in the crystal-so the normalisation yields the $A_{\mathrm{vib}}$ per unit cell. The phonon partition function $Z_{\mathrm{vib}}$ is computed from the (angular) phonon frequencies $\omega_{qj}$ according to:(9)$$Z_{\mathrm{vib}}\left(T\right)=\prod_{qj}\frac{\exp\left(-\hbar \omega_{qj}/2k_{B}T\right)}{1-\exp\left(-\hbar \omega_{qj}/k_{B}T\right)}$$
where the product runs over phonon modes with wavevector q and band index *j*. The vibrational internal energy $U_{\mathrm{vib}}$ can be computed according to:(10)$$U_{\mathrm{vib}}\left(T\right)=\sum_{qj}\left[n_{qj}\left(T\right)+\frac{1}{2}\right]\hbar \omega_{qj}=\sum_{qj}\left[\frac{1}{\exp\left(\hbar \omega_{qj}/k_{B}T\right)-1}+\frac{1}{2}\right]\hbar \omega_{qj}$$
where $n_{qj}$ is the phonon occupation number given by the Bose-Einstein distribution. The phonon entropy $S_{\mathrm{vib}}$ can then be obtained from $A_{\mathrm{vib}}$ and $U_{\mathrm{vib}}$.

Table 4 compares the calculated lattice energies and the 300 K $A_{\mathrm{vib}}$ of the five framework structures to bulk diamond Si, while Figure 5 compares the free energy differences ΔA of the five structures to d-Si as a function of temperature from 0–1000 K. The calculated lattice energies of the five framework structures are all higher than that of d-Si, indicating them to be metastable. For four of the five framework structures, viz. *o*C24, K-II/C-I, K-V/C-VI and K-VII/C-V, the differences in lattice energy $\Delta U_{\mathrm{latt}}$ fall within a relatively narrow range of 6.82–9.14 kJ mol${}^{-1}$ atom${}^{-1}$, which suggests these phases are energetically similar. On the other hand, C-II is predicted to have a ∼4 × larger $U_{\mathrm{latt}}$ than the other framework structures, which can be taken as a reflection of the weaker chemical bonding evident in the phonon spectra. Comparing the 300 K free-energy differences ΔA gives similar results. The differences in internal energy ΔU, including both lattice and phonon contributions, are comparable to the $\Delta U_{\mathrm{latt}}$, indicating that differences in vibrational internal energy are much smaller than differences in $U_{\mathrm{latt}}$. The differences in the vibrational entropy −TΔS are generally negative, i.e., the framework structures are stabilised by a higher entropy, but the stabilisation of 0.12–0.92 kJ mol${}^{-1}$ atom${}^{-1}$ is again much smaller than the differences in the lattice energies.

To further investigate the impact of phonon contributions to the free energy on the relative stabilities of the framework structures, we compare the varition in ΔA with temperture to the differences in internal energy ΔU and entropy −TΔS (Figure 6).

The ΔU all show slight increases with temperature, indicative of a larger $U_{\mathrm{vib}}$ that destabilises the framework materials relative to d-Si, but this effect is clearly secondary to the differences in the lattice internal energy. All five structures show a similar variation in internal energy with temperature, such that the ΔU between them remain roughly constant and the vibrational internal energy $U_{\mathrm{vib}}$ does not change the stability ordering predicted by the differences in the lattice energies. This is even the case for the orthorhombic *o*C24 and K-C/C-VI structures whose free energies are relatively close.

In contrast, differences in vibrational entropy favour the five framework structures relative to d-Si. For four of the five framework structures the stabilisation is <1 kJ mol${}^{-1}$ atom${}^{-1}$ at 1000 K, which is ∼5–10× smaller than the differences in the internal energy. For the C-II structure, on the other hand, the entropic stabilisation is much larger at around 3.5 kJ mol${}^{-1}$ atom${}^{-1}$. This can be attributed to the shift in the phonon spectrum to lower frequencies identified in Section 3.2 (c.f. Figure 3). However this does not compensate for the weaker bonding in this structure.

The study in Ref. [35] predicted the K-II/C-I, K-V/C-VI and K-VII/C-V structures to be 5.98, 7.62 and 5.31 kJ mol${}^{-1}$ atom${}^{-1}$ higher in energy than d-Si, respectively, which are comparable to our values of 7.47, 9.14 and 6.82 kJ mol${}^{-1}$ atom${}^{-1}$. On the other hand, Ref. [35] predicted the C-II structure to be 5.02 kJ mol${}^{-1}$ atom${}^{-1}$ higher in energy than bulk Si, which is a much smaller difference than predicted by our calculations and which would make this framework structure energetically competitive with the others. Similarly, our result is at odds with the computational study in Ref. [32], which predicted energy differences of 7.42 and 5.31 kJ mol${}^{-1}$ atom${}^{-1}$ with two different DFT functionals.

To investigate this further, we performed a series of additional calculations on d-Si and C-II to attempt to identify the source of the discrepancy (see Appendix A). These calculations indicated that the higher energy of the C-II structure predicted by our calculations is not due to our choice of functional or technical parameters, and indeed we obtained very similar results to those in the Materials Project entry (**mp-16220**) using the PBE functional including a calculated $\Delta U_{\mathrm{latt}}$ of 32.8 kJ mol${}^{-1}$ atom${}^{-1}$. In our view, the most likely explanation is that the ICSD structure reported with Ref. [32] (**ICSD: 56721**), which is listed as the source of **mp-16220**, is different to the structure(s) used in previous modelling studies. However, this is the only C-II structure available in either of the ICSD and MP database.

Despite this potential issue, our phonon calculations confirm that the C-II structure we investigate here is dynamically stable by the absence of imaginary modes in the calculated phonon spectrum (c.f. Figure 2b).

The empty Type-II structure has reportedly been synthesised by heating NaSi to form Na_8_Si_136_, followed by repeated acid washing and heating under vacuum to remove the Na guest atoms [37,67]. Heating a mixture of elemental Na and Si under pressure is reported to yield the filled Type-I structure, Na_8_Si_46_, and NaSi_6_, which can be converted to the empty *o*C24 by “thermal degassing” to remove the Na [36,55]. We would expect these preparation conditions potentially to yield metastable phases, and we therefore believe it feasible that the C-II structure examined here could in principle be formed. Indeed, as discussed in the following section, we predict a very low lattice thermal conductivity for this structure, which appears to be more in line with experimental measurements than predictions from previous computational studies [37,67,68,69], and which suggests this structure may be more representative of the phase obtained in experiments.

Given the K-V/C-VI and K-VII/C-V framework structures examined here are similar in energy to or lower in energy than the C-II, *o*C24 and K-II/C-I structures, which have (at least in principle) been prepared experimentally, this implies they should also be synthetically accessible given suitable synthesis conditions.

### 3.4. Thermal Transport

Using the single-mode relaxation-time approximation (SM-RTA) model, the macroscopic thermal conductivity $\kappa_{\mathrm{latt}}$ is computed as a sum of contributions from individual phonon modes as:(11)$$\kappa_{\mathrm{latt}}=\frac{1}{N}\sum_{qj}\kappa_{qj}\left(T\right)=\frac{1}{NV}\sum_{qj}C_{qj}\left(T\right)\nu_{qj}⊗\nu_{qj}\tau_{qj}\left(T\right)$$
Here *N* is the number of wavevectors in the summation and *V* is the unit-cell volume. The $C_{qj}$ are the modal heat capacities given by:(12)$$C_{qj}\left(T\right)=\sum_{qj}{\left[\frac{\hbar \omega_{qj}}{k_{B}T}\right]}^{2}\frac{\exp\left(\hbar \omega_{qj}/k_{B}T\right)}{{\left[\exp\left(\hbar \omega_{qj}/k_{B}T\right)-1\right]}^{2}}$$
The $\nu_{qj}$ are the mode group velocities given by the derivative of the frequencies $\omega_{qj}$ with respect to the wavevector:(13)$$\nu_{qj}=\frac{\partial \omega_{qj}}{\partial q}$$
The $\tau_{qj}$ are the phonon lifetimes, which are related to the phonon linewidths $\Gamma_{qj}$ by:(14)$$\tau_{qj}\left(T\right)=\frac{1}{\Gamma_{qj}\left(T\right)}$$
As some of the clathrates have non-cubic spacegroups and therefore show anisotropic thermal transport, we also compute a scalar average $\kappa_{\mathrm{ave}}$ according to:(15)$$\kappa_{\mathrm{ave}}=\frac{1}{3}\mathrm{Tr}\left(\kappa_{\mathrm{latt}}\right)=\frac{1}{3}\left(\kappa_{xx}+\kappa_{yy}+\kappa_{zz}\right)$$
where $\kappa_{xx}$, $\kappa_{yy}$ and $\kappa_{zz}$ are the three diagonal elements of the $\kappa_{\mathrm{latt}}$ tensor, corresponding respectively to transport along the principal *x*, *y* and *z* directions, and we have omitted the explicit temperature dependence for brevity. This average is comparable to what would be measured in experiments on e.g., thin films or pressed pellets consisting of randomly-oriented crystal grains.

The calculated $\kappa_{\mathrm{ave}}$ of bulk diamond Si is shown as a function of temperature in Figure 7 and compared to experimental measurements from Refs [30,31]. The SM-RTA model gives a near-quantitative reproduction of the experiments over a range of ∼150–1000 K with a mean absolute relative error of 8.46%. Over this temperature range the largest difference occurs at 1000 K, where the calculated value of 35.6 W m${}^{-1}$ K${}^{-1}$ overestimates the experimental values of 29.8 and 31 W m${}^{-1}$ K${}^{-1}$ reported in Refs [30,31] by 19.6 and 14.9% respectively. Given the high temperature, this could be due to limitations of the model, including not accounting for changes in volume due to thermal expansion at finite temperature and/or the omission of higher- (e.g., fourth-) order phonon scattering processes when computing the linewidths. However, we also note that there are significant differences in the two sets of experimental measurements at some temperatures—e.g., at 300 K Refs [30,31] report values of 142.2 and 156 W m${}^{-1}$ K${}^{-1}$, which differ from our calculated value of 136.2 W m${}^{-1}$ K${}^{-1}$ by 4.2 and 12.7%, respectively. These discrepancies highlight the fact that measuring thermal conductivity experimentally can be somewhat challenging.

Comparison of the calculated $\kappa_{\mathrm{ave}}$ of the six systems (Figure 8) shows a clear grouping whereby d-Si >*o*C24 ≃ K-II/C-I ≃ K-V/C-VI ≃ K-VII/C-V > C-II. At room temperature, the calculated $\kappa_{\mathrm{ave}}$ of d-Si and C-II are 136.2 and 6.33 W m${}^{-1}$ K${}^{-1}$, respectively, while the conductivities of the other four framework structures range from 30.45 W m${}^{-1}$ K${}^{-1}$ (K-VII/C-V) to 57.86 W m${}^{-1}$ K${}^{-1}$ (*o*C24) (Table 5). With reference to the phonon spectra (Figure 2 and Figure 3), the high thermal conductivity of d-Si can be explained by the wide dispersion of the acoustic modes and consequent large group velocities $\nu_{qj}$, and the low thermal conductivity of the C-II structure could in principle be explained by its generally lower phonon frequencies, weaker chemical bonding, and resulting smaller group velocities.

Among the other four framework systems the $\kappa_{\mathrm{ave}}$ fall in the order of *o*C24 > K-II/C-I > K-V/C-VI > K-VII/C-V. This can be largely explained by the fact that high thermal conductivity is generally associated with less complex structures (i.e., with smaller primitive cells) and high symmetry: the *o*C24 and K-VII/C-V structures have the smallest and largest cells, respectively, while the K-II/C-I and K-V/C-VI structures have similar-sized primitive cells but K-II/C-I is cubic whereas K-V/C-VI is othorhombic (c.f. Table 2).

To better understand the differences in the thermal transport of the six systems we use the constant relaxation-time approximation (CRTA) model developed in Refs [18,39], whereby the $\kappa_{\mathrm{latt}}$ is written as the product of a harmonic term and a weighted-average lifetime $\tau^{\mathrm{CRTA}}$:(16)$$\kappa_{\mathrm{latt}}≃\tau^{\mathrm{CRTA}}\times \frac{1}{N}\sum_{qj}\frac{\kappa_{qj}}{\tau_{qj}}=\tau^{\mathrm{CRTA}}\times \frac{1}{NV}\sum_{qj}C_{qj}\nu_{qj}⊗\nu_{qj}$$
where we have again omitted the temperature dependence for brevity. Comparing the harmonic term in the summand and the $\tau^{\mathrm{CRTA}}$ allows the difference in $\kappa_{\mathrm{latt}}$ between systems to be attributed quantitatively to differences in the group velocities and lifetimes. We note that both the harmonic and lifetime terms are implicitly temperature dependent. While in this analysis we follow Ref. [39] and treat $\tau^{\mathrm{CRTA}}$ as a scalar quantity, the harmonic term is a tensor, and we therefore compute an equivalent scalar average to that in Equation (Equation 15) in order to analyse the $\kappa_{\mathrm{ave}}$.

The harmonoic and lifetime terms of the six systems as a function of temperature are compared in Figure 8b and Figure 8c respectively, and values at 300 K are included in Table 5. This analysis clearly shows that the differences in both the harmonic and lifetime terms are responsible for the variation in the $\kappa_{\mathrm{ave}}$. At 300 K the harmonic term for bulk Si is 5 W m${}^{-1}$ K${}^{-1}$ ps${}^{-1}$, which is the largest of the six compounds and can, as noted above, be explained by the wide dispersion of the acoustic modes shown in Figure 2. The *o*C24 structure has the second-highest $\kappa /\tau^{\mathrm{CRTA}}$ at 2.3 W m${}^{-1}$ K${}^{-1}$ ps${}^{-1}$ (54% smaller), and the remaining framework structures all have $\kappa /\tau^{\mathrm{CRTA}}$ < 1 W m${}^{-1}$ K${}^{-1}$ ps${}^{-1}$ (83–91% smaller). This ordering shows that the harmonic term is primarily governed by the size of the primitive cells such that larger cells result in lower $\kappa /\tau^{\mathrm{CRTA}}$.

The variation in the lifetime term is more intricate. At 300 K, d-Si has a $\tau^{\mathrm{CRTA}}$ of 27.2 ps. The *o*C24 framework has a 35% smaller averaged lifetime of 17.8 ps which, in addition to its smaller harmonic term, results in its 70% lower $\kappa_{\mathrm{ave}}$. On the other hand, the K-II/C-I, K-V/C-VI and K-VII/C-V structures have averaged lifetimes that are ∼1.5–2× longer than that of d-Si, which partially compensates for their smaller $\kappa /\tau^{\mathrm{CRTA}}$.

The low thermal conductivity of the C-II structure, on the other hand, appears to be characterised by both a low $\kappa /\tau^{\mathrm{CRTA}}$ and a short $\tau^{\mathrm{CRTA}}$. The former is the smallest among the six systems, at ∼10% of the value for d-Si, which can be explained by the impact of the weaker chemical bonding on the group velocities. The $\tau^{\mathrm{CRTA}}$ is also the shortest of the six compounds and is around 50% smaller than that of d-Si.

The phonon lifetimes are the inverse of the phonon linewidths $\Gamma_{qj}$ (c.f. Equation (Equation 14)). Within the SM-RTA model the $\Gamma_{qj}$ are calculated as a sum of contributions from three-phonon scattering processes between triplets of modes qj, $q^{\prime }j^{\prime }$ and $q^{\prime \prime }j^{\prime \prime }$ according to: [50]
(17)$$\begin{array}{cc} \Gamma_{qj}\left(T\right)= & \frac{18\pi }{\hbar^{2}}\sum_{q^{\prime }j^{\prime },q^{\prime \prime }j^{\prime \prime }}{\left|\Phi_{-qj,q^{\prime }j^{\prime },q^{\prime \prime }j^{\prime \prime }}\right|}^{2}\times \\ \; & \{\left[n_{q^{\prime }j^{\prime }}\left(T\right)-n_{q^{\prime \prime }j^{\prime \prime }}\left(T\right)\right]\left[\delta \left(\omega_{qj}+\omega_{q^{\prime }j^{\prime }}-\omega_{q^{\prime \prime }j^{\prime \prime }}\right)-\delta \left(\omega_{qj}-\omega_{q^{\prime }j^{\prime }}+\omega_{q^{\prime \prime }j^{\prime \prime }}\right)\right]+\\ \; & \left[n_{q^{\prime }j^{\prime }}\left(T\right)+n_{q^{\prime \prime }j^{\prime \prime }}\left(T\right)+1\right]\delta \left(\omega_{qj}-\omega_{q^{\prime }j^{\prime }}-\omega_{q^{\prime \prime }j^{\prime \prime }}\right)\} \end{array}$$
The contribution from each triplet is determined by the three-phonon interaction strengths $\Phi_{-qj,q^{\prime }j^{\prime },q^{\prime \prime }j^{\prime \prime }}$, the occupation numbers $n_{qj}$, and selection rules enforcing the conservation of energy. The two terms in the brace in the summation describe two types of scattering process, viz. collisions (Class 1), where two phonons coalesce to form a third, and decay (Class 2), where a phonon decays and emits two others. (There is also a selection rule for the conservation of (crystal) momentum, which is incorpared into the definition of the $\Phi_{-qj,q^{\prime }j^{\prime },q^{\prime \prime }j^{\prime \prime }}$ [50]).

The three-phonon interaction strengths are temperature independent and calculated from the harmonic frequencies and eigenvectors $\omega_{qj}$/$W_{qj}$ and the third-order (anharmonic) force constants $\Phi^{\left(3\right)}$, while the conservation of energy expression is calculated solely from the $\omega_{qj}$ and is temperature dependant through the $n_{qj}$. Based on the analysis in Ref. [50], we approximate the linewidths as:(18)$$\Gamma_{qj}\left(T\right)≃\frac{18\pi }{\hbar^{2}}\tilde{P}N_{2}\left(q_{j},\omega_{qj},T\right)$$
where $\tilde{P}$ is a weighted squared average three-phonon interaction strength and $N_{2}\left(q,\omega ,T\right)$ is a weighted two-phonon joint density of states (w-JDoS) defined as:(19)$$N_{2}\left(q,\omega ,T\right)=N_{2}^{\left(1\right)}\left(q,\omega ,T\right)+N_{2}^{\left(2\right)}\left(q,\omega ,T\right)$$
with:(20)$$\begin{array}{cc} N_{2}^{\left(1\right)}\left(q,\omega ,T\right)= & \frac{1}{N}\sum_{q^{\prime }j^{\prime },q^{\prime \prime }j^{\prime \prime }}\Delta \left(-q+q^{\prime }+q^{\prime \prime }\right)\left[n_{q^{\prime }j^{\prime }}\left(T\right)-n_{q^{\prime \prime }j^{\prime \prime }}\left(T\right)\right]\times \\ \; & \left[\delta \left(\omega +\omega_{q^{\prime }j^{\prime }}-\omega_{q^{\prime \prime }j^{\prime \prime }}\right)-\delta \left(\omega -\omega_{q^{\prime }j^{\prime }}+\omega_{q^{\prime \prime }j^{\prime \prime }}\right)\right] \end{array}$$
(21)$$\begin{array}{cc} N_{2}^{\left(2\right)}\left(q,\omega ,T\right)= & \frac{1}{N}\sum_{q^{\prime }j^{\prime },q^{\prime \prime }j^{\prime \prime }}\Delta \left(-q+q^{\prime }+q^{\prime \prime }\right)\times \\ \; & \left[n_{q^{\prime }j^{\prime }}\left(T\right)+n_{q^{\prime \prime }j^{\prime \prime }}\left(T\right)+1\right]\delta \left(\omega -\omega_{q^{\prime }j^{\prime }}-\omega_{q^{\prime \prime }j^{\prime \prime }}\right) \end{array}$$
For a given mode with a wavevector q and frequency ω these functions count respectively the number of possible energy- and momentum-conserving Class 1 and Class 2 events.

As per Equation (Equation 11) the $\kappa_{\mathrm{latt}}$ is proportional to the $\tau_{qj}$ and therefore inversely proportional to the $\Gamma_{qj}$, so that when the squared three-phonon interaction strengths are replaced by a constant $\tilde{P}$ the $\kappa_{\mathrm{latt}}$ is proportional to $1/\tilde{P}$. A suitable $\tilde{P}$ to reproduce the scalar average $\kappa_{\mathrm{latt}}$ can therefore be obtained from a linear fit of the $\kappa_{\mathrm{latt}}$ as a function of $1/\tilde{P}$. As noted in Ref. [39], although the $\Phi_{-qj,q^{\prime }j^{\prime },q^{\prime \prime }j^{\prime \prime }}$ in Equation (Equation 17) are temperature independent, replacing them with a single value cannot in general capture the temperature dependence of the $\kappa_{\mathrm{latt}}$, so the $\tilde{P}$ are weakly temperature dependent and the fit must be performed at a chosen temperature *T*.

Both the $\tilde{P}$ and $N_{2}\left(q,\omega ,T\right)$ can be used to compare between materials and attribute differences in the phonon linewidths/lifetimes to differences in the phonon interaction strengths, which are anharmonic by definition, and to differences in the numbers of allowed scattering pathways dictated by differences in the harmonic phonon spectra. To compare the $N_{2}\left(q,\omega ,T\right)$ it is prudent to average over q to obtain a function of frequency only, i.e.,:(22)$$\begin{array}{cc} {\bar{N}}_{2}\left(\omega ,T\right) & ={\bar{N}}_{2}^{\left(1\right)}\left(\omega ,T\right)+{\bar{N}}_{2}^{\left(2\right)}\left(\omega ,T\right)\\ \; & =\frac{1}{N}\sum_{q}N_{2}^{\left(1\right)}\left(q,\omega ,T\right)+\frac{1}{N}\sum_{q}N_{2}^{\left(2\right)}\left(q,\omega ,T\right) \end{array}$$
We note that both the $N_{2}\left(q,\omega ,T\right)$/${\bar{N}}_{2}\left(\omega ,T\right)$ and $\tilde{P}$ are “extensive”, i.e., they scale with the number of atoms in the primitive cell, and that to compare between materials the w-JDoS must be scaled down by ${\left(3n_{a}\right)}^{2}$ and the $\tilde{P}$ scaled up by ${\left(3n_{a}\right)}^{2}$.

The scaled ${\bar{N}}_{2}\left(\omega \right)$ of the six structures at *T* = 300 K are compared in Figure 9a. The w-JDoS of all of the structures apart from C-II have a very similar shape, particular at lower frequencies, suggesting there are similar numbers of allowed scattering pathways. On the other hand, the ${\bar{N}}_{2}\left(\omega \right)$ of C-II indicates that there are a considerably larger number of scattering pathways available across most of its frequency spectrum. The shape of the w-JDoS depends on the shape of the harmonic phonon spectrum, and this therefore reflects the different distribution of frequencies in the C-II spectrum compared to the other structures shown in Figure 3.

A value of $\tilde{P}$ was fitted for each of the six structures at *T* = 300 K (see Appendix A), and the scaled values are listed alongside the $\kappa_{\mathrm{latt}}$ and CRTA components in Table 5. Excluding C-II, a plot of $1/\tilde{P}$ against the averaged lifetime $\tau^{\mathrm{CRTA}}$ (Figure 9b) shows a strong linear correlation across all the framework structures ($R^{2}$ = 0.997), providing clear evidence that differences in the $\tau^{\mathrm{CRTA}}$ can predominantly be attributed to differences in the phonon-phonon interaction strengths. The $\tilde{P}$ fitted for the C-II structure lie slightly below the trendline, i.e., its lifetimes are lower than would be expected considering only the strength of the phonon-phonon interactions, and this can be accounted for by its generally larger ${\bar{N}}_{2}\left(\omega \right)$ (c.f. Figure 9a).

We therefore attribute the variation in the phonon lifetimes in the framework structures relative to bulk diamond Si predominantly to differences in the phonon-phonon interaction strengths, and the short $\tau_{qj}$ of the C-II structure to a combination of strong phonon interactions plus a larger number of allowed scattering pathways. This conclusion is in line with the modelling study in Ref. [69], which also highlighted the importance of phonon-phonon interaction strengths in determining the thermal conductivities of d-Si and the Si clatrates.

Our calculated room-temperature $\kappa_{\mathrm{ave}}$ of 6.33 W m${}^{-1}$ K${}^{-1}$ is of a similar magnitude to the value of 2.5 W m${}^{-1}$ K${}^{-1}$ reported for the empty Type-II clathrate, Ref. [37] and to the values of ∼2.5 and 1 W m${}^{-1}$ K${}^{-1}$ reported for Na_1_Si_136_ and Na_8_Si_136_ respectively [67]. However, our value is considerably smaller than the values of 52 and 57.5 W m${}^{-1}$ K${}^{-1}$ obtained from calculations similar to those in the present work. [68,69] On the other hand, the study in Ref. [69] obtained values of 151 and 53 W m${}^{-1}$ K${}^{-1}$ for d-Si and the Type-I clathrate structure, which are much more comparable to, albeit slightly higher than, our predicted values. Similarly, the study in Ref. [68] predicted a $\kappa_{\mathrm{latt}}$ on the order of 100 W m${}^{-1}$ K${}^{-1}$ for bulk Si, which is again comparable to the results obtained in the present work.

To try to elucidate the origin of the discrepancy with previous modelling studies we performed some additional calculations of the phonon dispersion and $\kappa_{\mathrm{latt}}$ (see Appendix A), based on which we again concluded that the most likely explanation is the C-II structure reported with Ref. [32] being different to the structure(s) used in the modelling studies in Refs [68,69]. On the face of it, our calculated $\kappa_{\mathrm{latt}}$ is much more in keeping with the low thermal conductivity measured experimentally [37,67]. However, the low experimental values may be due in part to high concentrations of structural defects, which are difficult to avoid given the harsh processing conditions required to remove the Na guest atoms, and indeed the low density of the Na-filled samples in the study in Ref. [67] may be indicative of similar issues in these measured thermal conductivities. Although these issues were raised in Ref. [69] as a possible reason for the large difference between the calculations and the measurements in Ref. [37], it is debatable whether they could account for the ∼20× difference.

Finally, we also consider the anisotropy in the thermal conductivities of the three framework structures with non-cubic spacegroups. Figure 10 shows the temperature dependence of the $\kappa_{xx}$, $\kappa_{yy}$ and $\kappa_{zz}$ components of the $\kappa_{\mathrm{latt}}$ tensors in the *o*C24, K-V/C-VI and K-VII/C-V frameworks. For the orthorhombic *o*C24 and K-V/C-VI structures the three $\kappa_{\mathrm{latt}}$ components correspond to transport along the crystallographic *a*, *b* and *c* axes. The *o*C24 structure shows relatively large anisotropy. The largest $\kappa_{xx}$ = 54.86 W m${}^{-1}$ K${}^{-1}$ is along the shortest crystallographic *a* axis, which makes sense as the short lattice constant of 3.826 Å would enforce a high degree of regularity in the bonding along this direction (c.f. Table 2). On the other hand, the *b* and *c* axes have similar lengths of *b* = 10.697 and 12.660 Å, respectively, but the corresponding $\kappa_{\mathrm{latt}}$ differ by a factor of two. The anisotropy in the K-V/C-VI structure is much smaller, with the $\kappa_{\mathrm{latt}}$ components ranging from $\kappa_{xx}$ = 38.24 W m${}^{-1}$ K${}^{-1}$ to $\kappa_{zz}$ = 32.96 W m${}^{-1}$ K${}^{-1}$. Although the largest transport is again associated with the shortest *a* axis, the transport along the longest *b* axis is similar and the smallest $\kappa_{\mathrm{latt}}$ occurs along the *c* axis, which has a similar length to the *a* axis. Finally, the K-VII/C-V system has a hexagonal spacegroup in which the $\kappa_{xx}$ and $\kappa_{yy}$ components are equal and smaller than the $\kappa_{zz}$ component, corresponding to transport along the longer *c* axis, by around 2 W m${}^{-1}$ K${}^{-1}$.

## 4. Discussion

In this work we have performed a detailed study of the structural dynamics, energetics and thermal transport of bulk diamond Si and five Si framework structures including four clathrates.

From the calculated phonon spectra, we find that four of the five framework structures show a similar distribution of phonon frequencies to bulk Si, despite their more complex dispersions, whereas the spectrum of the Clathrate II structure is notably skewed towards lower frequencies. We have provided reference infrared and Raman spectra for each of the six systems, comparison of which indicates that it should be possible to use a combination of both techniques to establish or confirm the identity of phases prepared in experiments.

The energy differences predicted by the calculations indicate that all five framework structures are metastable with respect to bulk Si. For the majority of the frameworks the energy differences are on the order of 5–10 kJ mol${}^{-1}$ atom${}^{-1}$, but the C-II phase is predicted to be much less stable at >30 kJ mol${}^{-1}$ atom${}^{-1}$ above d-Si. While the framework structures, and in particular C-II, are stabilised at elevated temperature due to their higher vibrational entropy, this is insufficient to compensate for differences in the lattice energies and therefore does not affect the energetic ordering. Although the predicted high energy of the C-II phase is at odds with other computational studies, the calculated phonon dispersion indicates it to be dynamically stable, and the experimental synthesis technique employed to form it would in principle be capable of yielding such high-energy metastable phases.

All of the framework structures are predicted to have substantially smaller lattice thermal conductivity than d-Si, which can be attributed quantitatively to a reduction in the phonon group velocities and an intricate variation in the phonon lifetimes. The differences in group velocity correlate to the size and, to a lesser extent, symmetry of the primitive cells. The variation in the phonon lifetimes is predominantly due to differences in the strength of the interactions between phonon modes, indicting that the different structures show different levels of phonon anharmonicity.

Notably, we predict a very low room-temperature $\kappa_{\mathrm{latt}}$ of 6.33 W m${}^{-1}$ K${}^{-1}$ for the C-II structure, which is an order of magnitude smaller than predicted by two previous computational studies but appears to be a good match to experimental measurements. The limited heat transport in this framework structure can be attributed to low group velocities, reflecting weak bonding evident in its phonon spectrum, and to short lifetimes arising from a combination of strong phonon interactions and an increased number of allowed three-phonon scattering processes compared to the other systems.

Overall, this study highlights the utility of first-principles calculations for studying the structural dynamics and heat transport in complex framework structures, and in particular for establishing the origin of differences in lattice thermal conductivity. The latter is of particular interest in the field of thermoelectric materials, as it may provide valuable guidance on appropriate materials engineering strategies for optimising the $\kappa_{\mathrm{latt}}$. In this direction, the low $\kappa_{\mathrm{latt}}$ of the C-II structure predicted in this study and obtained in experiments suggests that further studies on this phase are warranted and may prove fruitful.

## Figures and Tables

**Figure 1 molecules-27-06431-f001:**
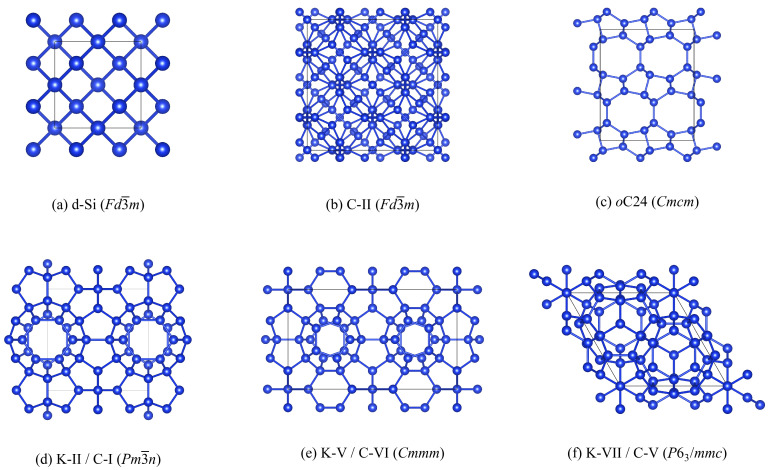
Images of the six structures examined in this work: (**a**) diamond Si (d-Si); (**b**) the Type-II clathrate (C-II); (**c**) the “*o*C24” structure reported in Ref. [36]; (**d**) the Type-I clathrate reported as ${\mathrm{Si}}_{K}^{\mathrm{II}}$ in Ref. [35] (K-II/C-I); (**e**) the ${\mathrm{Si}}_{K}^{V}$ structure reported in Ref. [35], which is similar to the Type-VI clathrate [34] (K-V/C-VI); and (**f**) the Type-V clathrate identified as ${\mathrm{Si}}_{K}^{\mathrm{VII}}$ in Ref. [34] (K-VII/C-V). These images were prepared using the VESTA software [65].

**Figure 2 molecules-27-06431-f002:**
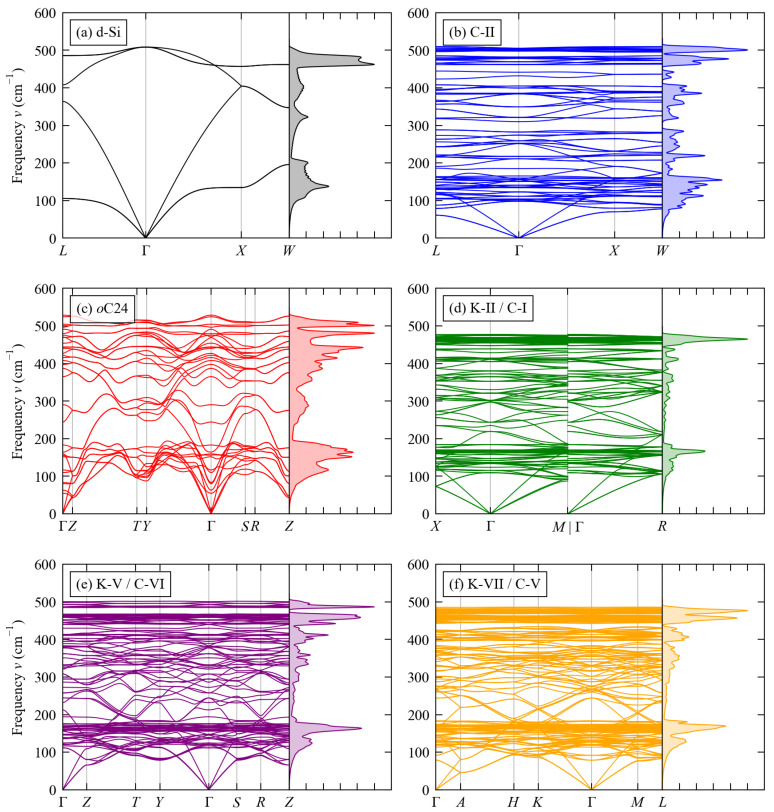
Calculated phonon dispersion ν(q) and density of states g(ν) curves of the six structures examined in this work: (**a**) d-Si, (**b**) C-II, (**c**) *o*C24, (**d**) K-II/C-I, (**e**) K-V/C-VI, and (**f**) K-VII/C-V.

**Figure 3 molecules-27-06431-f003:**
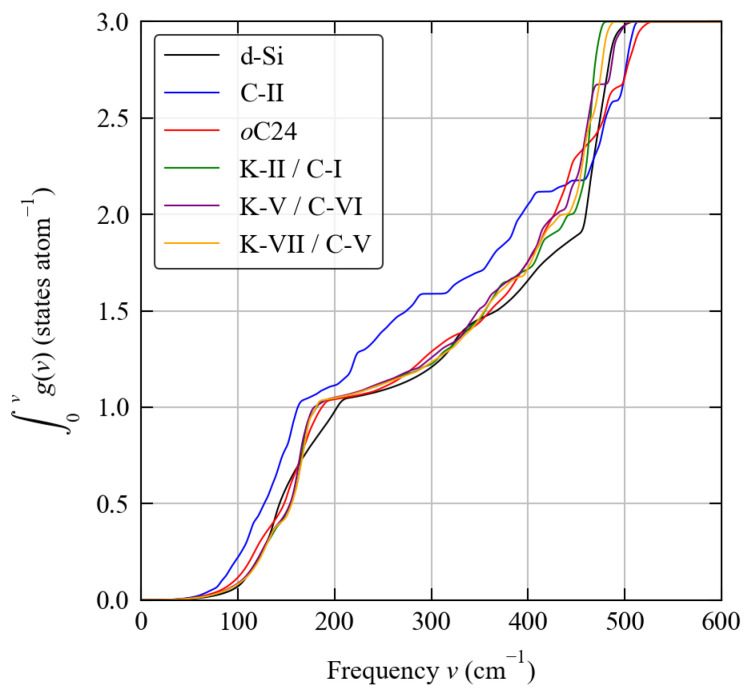
Cumulative integral of the phonon density of states curves g(ν) in Figure 2 as a function of frequency showing the distributions of modes over frequencies.

**Figure 4 molecules-27-06431-f004:**
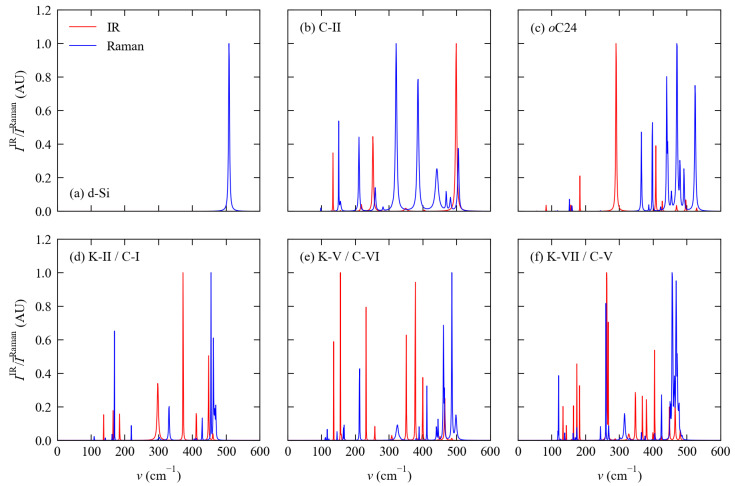
Simulated infrared (IR; red) and Raman (blue) spectra of the six structures examined in this work obtained using the method in Ref. [52]: (**a**) d-Si, (**b**) C-II, (**c**) *o*C24, (**d**) K-II/C-I, (**e**) K-V/C-VI, and (**f**) K-VII/C-V. The Raman activities used to generate these spectra are the scalar averages ${\bar{I}}^{\mathrm{Raman}}$ defined in Equation (Equation 6).

**Figure 5 molecules-27-06431-f005:**
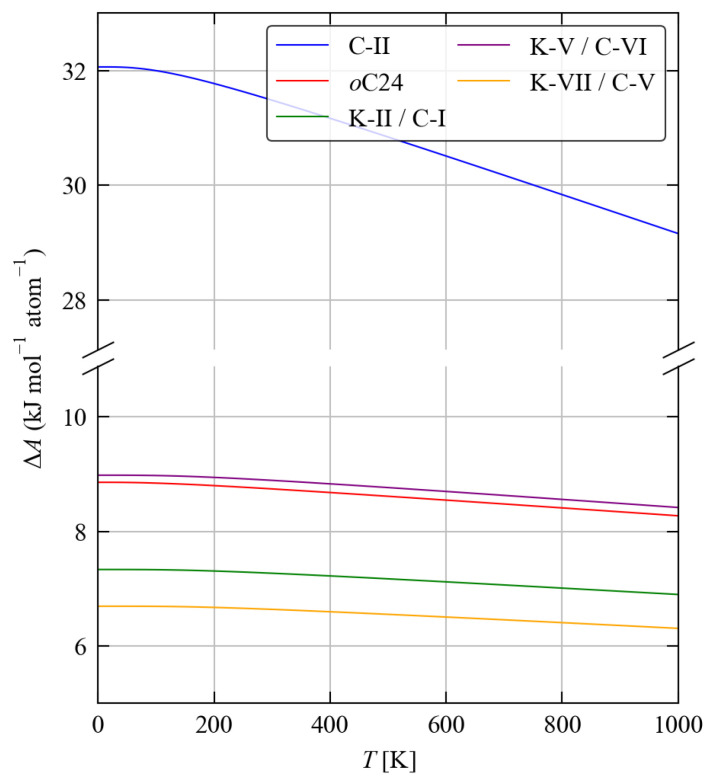
Calculated Helmholtz free energy differences $\Delta A=\Delta U_{\mathrm{latt}}+\Delta U_{\mathrm{vib}}-T\Delta S_{\mathrm{vib}}$ as a function of temperature for the five Si framework structures relative to bulk diamond Si. Note the break in the *y* axis.

**Figure 6 molecules-27-06431-f006:**
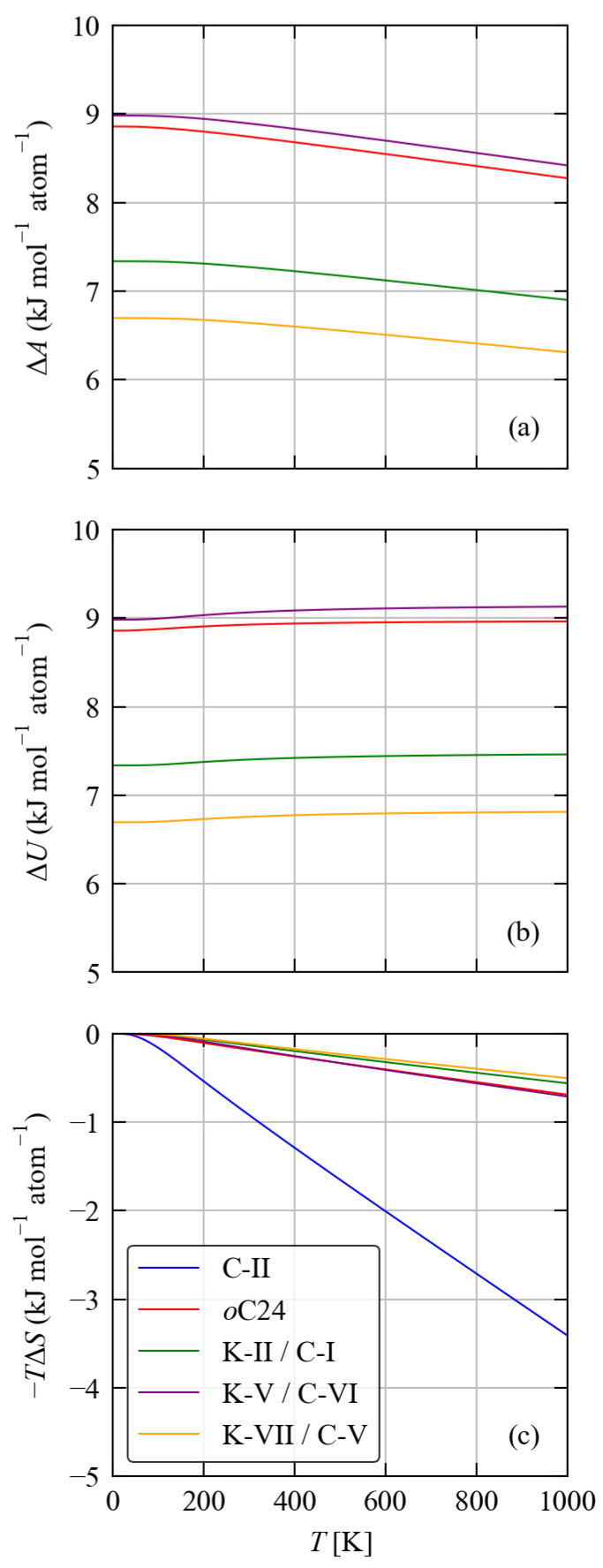
Breakdown of the Helmholtz free energy differences ΔA (**a**), shown in Figure 5, into differences in the internal energy $\Delta U=\Delta U_{\mathrm{latt}}+\Delta U_{\mathrm{vib}}$ (**b**) and entropy $-T\Delta S_{\mathrm{vib}}$ (**c**).

**Figure 7 molecules-27-06431-f007:**
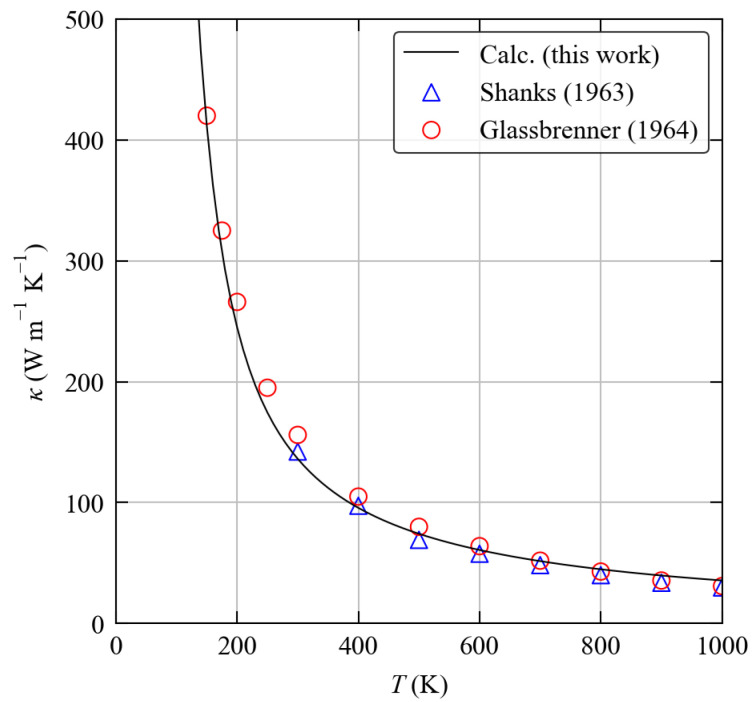
Calculated averaged lattice thermal conductivity of bulk diamond Si as a function of temperature (black line) compared to experimental data from Refs [30,31] (blue triangles/red circles).

**Figure 8 molecules-27-06431-f008:**
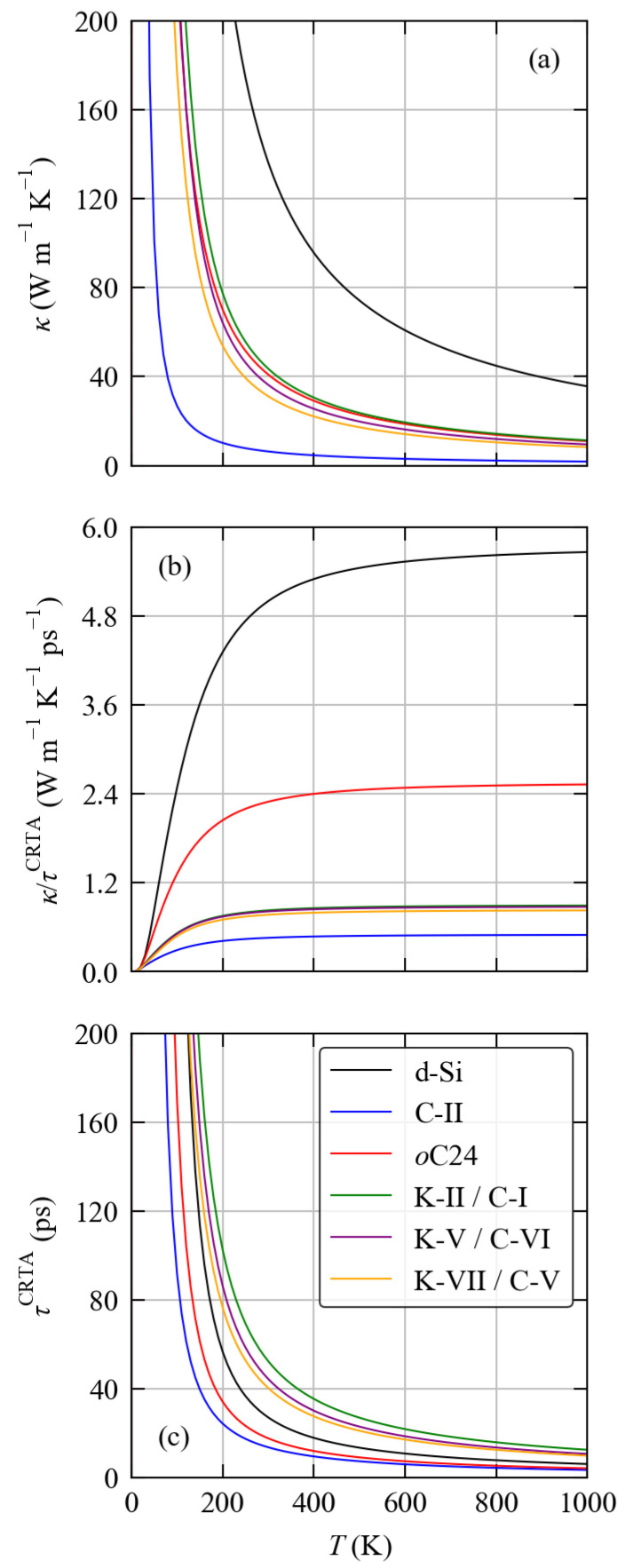
Calculated averaged lattice thermal conductivity of the six structures examined in this work: d-Si (black), C-II (blue), *o*C24 (red), K-II/C-I (green), K-V/C-VI (purple) and K-VII/C-V (orange). Subplot (**a**) shows the κ as a function of temperature, and subplots (**b**,**c**) show its decomposition into a harmonic term $\kappa /\tau^{\mathrm{CRTA}}$ and averaged lifetime $\tau^{\mathrm{CRTA}}$ using the constant-relaxation time approximation (CRTA) model defined in Equation (Equation 16) in the text.

**Figure 9 molecules-27-06431-f009:**
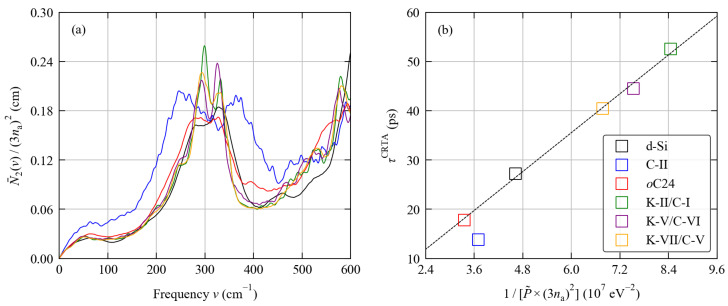
Analysis of the phonon lifetimes of the six systems examined in this work using the model defined in Equation (Equation 18) in the text. Plot (**a**) compares the scaled weighted two-phonon joint density of states (w-JDoS) functions ${\bar{N}}_{2}\left(\omega \right)$ at *T* = 300 K (Equation (Equation 22)). Plot (**b**) compares the averaged lifetimes $\tau^{CRTA}$ defined in Equation (Equation 16) to the inverse averaged squared phonon-phonon interaction strengths $1/\tilde{P}$. The line colours used in Plot (**a**) match the marker colours in Plot (**b**). In Plot (**b**) the dashed black line is a linear fit to the data for d-Si and the *o*C24, K-II/C-I, K-V/C-VI and K-VII/C-V framework structures.

**Figure 10 molecules-27-06431-f010:**
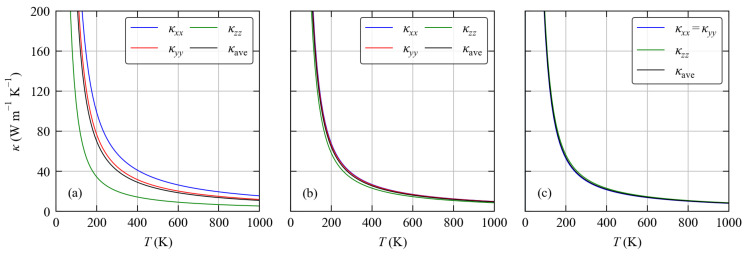
Anisotropy in the lattice thermal conductivity of the *o*C24 (**a**), K-V/C-VI (**b**) and K-VII/C-V (**c**) structures. For the orthorhombic structures shown in (**a**,**b**) the three diagonal components of the $\kappa_{\mathrm{latt}}$ tensor, $\kappa_{xx}$, $\kappa_{yy}$ and $\kappa_{zz}$ are shown together with the average $\kappa_{\mathrm{ave}}$ defined in Equation (Equation 15). For the hexagonal K-VII/C-V structure shown in (**c**) the $\kappa_{xx}$ and $\kappa_{yy}$ are equivalent, and $\kappa_{xx}=\kappa_{yy}$ is shown together with $\kappa_{zz}$ and $\kappa_{\mathrm{ave}}$.

**Table 1 molecules-27-06431-t001:** Summary of the technical parameters used in calculations on the six structures examined in this work: unit-cell definition and k-point sampling used for geometry optimisation and to calculate the infrared (IR) and Raman activities; cell definition, supercells and k-point sampling used to compute the second- and third-order force constants $\Phi^{\left(2\right)}$/$\Phi^{\left(3\right)}$; and q-point meshes used to compute the phonon density of states (DoS) and vibrational Helmholtz free energy $A_{\mathrm{vib}}$ and to evaluate the lattice thermal conductivity $\kappa_{\mathrm{latt}}$. Cell definitions are either the primitive (“Prim.”) or conventional (“Conv.”) cells, for systems where these differ. k- and q-point meshes are given as regular Γ-centered meshes generated using the Monkhorst-Pack scheme [47].

		d-Si	C-II	*o*C24	K-II/C-I	K-V/C-VI	K-VII/C-V
Optimisation	Cell	Conv.	Prim.	Prim.	-	Prim.	-
	k-points	6 × 6 × 6	4 × 4 × 4	8 × 8 × 2	3 × 3 × 3	4 × 4 × 3	4 × 4 × 2
$I_{\mathrm{IR}}$/$I_{\mathrm{Raman}}$	Cell	Prim.	Prim.	Prim.	-	Prim.	-
	k-points	10 × 10 × 10	4 × 4 × 4	8 × 8 × 2	3 × 3 × 3	4 × 4 × 3	4 × 4 × 2
$\Phi^{\left(2\right)}$	Cell	Conv.	Conv.	Conv.	-	Conv.	-
	Supercell (# atoms) ^1^	3 × 3 × 3 (216)	$\left(\begin{array}{ccc} 0 & 1 & 1\\ 1 & 0 & 1\\ 1 & 1 & 0 \end{array}\right)$ (272)	$\left(\begin{array}{ccc} 2 & 2 & 0\\ -2 & 2 & 0\\ 0 & 0 & 2 \end{array}\right)$ (384)	2 × 2 × 2 (368)	$\left(\begin{array}{ccc} 1 & 1 & 0\\ -1 & 1 & 0\\ 0 & 0 & 2 \end{array}\right)$ (320)	2 × 2 × 1 (272)
	k-points	2 × 2 × 2	2 × 2 × 2	2 × 2 × 1	2 × 2 × 2	2 × 2 × 2	2 × 2 × 2
$\Phi^{\left(3\right)}$	Cell	Conv.	Conv.	Conv.	-	Conv.	-
	Supercell (# atoms)	2 × 2 × 2 (64)	1 × 1 × 1 (136)	3 × 1 × 1 (72)	1 × 1 × 1 (46)	1 × 1 × 1 (80)	1 × 1 × 1 (68)
	k-points	3 × 3 × 3	2 × 2 × 2	3 × 2 × 2	3 × 3 × 3	4 × 2 × 3	4 × 4 × 2
q-points	DoS/$A_{\mathrm{vib}}$	48 × 48 × 48	24 × 24 × 24	40 × 40 × 16	24 × 24 × 24	24 × 24 × 24	24 × 24 × 16
	$\kappa_{\mathrm{latt}}$ ^2^	36 × 36 × 36	16 × 16 × 16	13 × 13 × 5	13 × 13 × 13	9 × 9 × 9	14 × 14 × 9

^1^ The non-diagonal supercell matrices used for C-II, *o*C24 and K-V/C-VI are equivalent to 2 × 2 × 2, 4 × 4 × 2 and 2 × 2 × 2 expansions of the respective primitive cells. ^2^ Convergence of the $\kappa_{\mathrm{latt}}$ of C-II with respect to the q-point sampling density is slow, such that a coarser 8 × 8 × 8 mesh yields a $\kappa_{\mathrm{latt}}$ within ~5% of the converged value obtained with the 24 × 24 × 24 mesh used to compute the density of states and vibrational free energy.

**Table 2 molecules-27-06431-t002:** List of the six structures examined in this study with the sources of the initial structures (mp—Materials Project; ICSD—Inorganic Crystal Structure Database), the crystallographic spacegroups, and the number of atoms, lattice constants (*a*, *b*, *c*) and volumes *V* of the conventional unit cells.

	Source	Spacegroup	$n_{a}$	*a* (Å)	*b* (Å)	*c* (Å)	*V* (Å^3^)
d-Si	**mp-149**	$Fd\bar{3}m$	8	5.436	-	-	160.6
C-II	**mp-16220** [32]	$Fd\bar{3}m$	136	14.629	-	-	3130.9
*o*C24	**mp-1095269** [36]	Cmcm	24	3.826	10.697	12.660	518.1
K-II/C-I	**mp-971662** [35]	$Pm\bar{3}m$	46	10.227	-	-	1052.5
K-V/C-VI	**ICSD: 189396** [35]	Cmmm	80	10.158	17.596	10.325	1845.6
K-VII/C-V	**mp-1203790** [35]	$P6_{3}/mmc$	68	10.358	-	16.956	1575.3

**Table 3 molecules-27-06431-t003:** Analysis of the 3N
Γ-point phonon modes in the six structures examined in this work. For each structure we list the irreducible representation $\Gamma_{3N}$ spanned by the modes together with the expected numbers of infrared (IR)- and Raman-active modes based on symmetry.

	3N	$\Gamma_{3N}$	# IR Active	# Raman Active
d-Si	6	T1u + T2g	1	1
C-II	102	3 A1g + A1u + A2g + 3 A2u + 4 Eg + 4 Eu + 5 T1g + 8 T1u + 8 T2g + 5 T2u	8	15
*o*C24	36	6 Ag + 3 Au + 3 B1g + 6 B1u + 3 B2g + 6 B2u + 6 B3g + 3 B3u	15	18
K-II/C-I	138	3 A1g + 2 A1u + 4 A2g + 2 A2u + 7 Eg + 4 Eu + 8 T1g + 10 T1u + 8 T2g + 9 T2u	10	18
K-V/C-VI	120	18 Ag + 11 Au + 15 B1g + 16 B1u + 13 B2g + 17 B2u + 14 B3g + 16 B3u	49	50
K-VII/C-V	204	11 A1g + 5 A1u + 7 A2g + 11 A2u + 5 B1g + 11 B1u + 11 B2g + 7 B2u + 16 E1g + 18 E1u + 18 E2g + 16 E2u	29	45

**Table 4 molecules-27-06431-t004:** Relative energies of the five framework structures examined in this study compared to bulk diamond Si. For each system we show the differences in the lattice internal energies $U_{\mathrm{latt}}$ together with the differences in the Helmholtz energies A=U−TS at *T* = 300 K and the corresponding differences in the internal energy $U=U_{\mathrm{latt}}+U_{\mathrm{vib}}$ and entropy −TS.

	Δ (kJ mol^−1^ atom^−1^)			
	$\Delta U_{\mathrm{latt}}$	ΔA	ΔU	−TΔS
d-Si	-	-	-	-
C-II	32.62	31.47	32.39	−0.92
*o*C24	8.97	8.73	8.91	−0.18
K-II/C-I	7.47	7.25	7.39	−0.13
K-V/C-VI	9.14	8.88	2.70	−0.17
K-VII/C-V	6.82	6.62	6.74	−0.12

**Table 5 molecules-27-06431-t005:** Calculated lattice thermal conductivity of the six structures examined in this work at *T* = 300 K. For each structure the unique diagonal components $\kappa_{xx}$, $\kappa_{yy}$ and $\kappa_{zz}$ of the $\kappa_{\mathrm{latt}}$ tensor, corresponding to transport along the Cartesian *x*, *y* and *z* directions, are shown together with the average $\kappa_{\mathrm{ave}}$ defined in Equation (Equation 15). Decompositions of the $\kappa_{\mathrm{ave}}$ into a harmonic term $\kappa /\tau^{\mathrm{CRTA}}$ and lifetime term $\tau^{\mathrm{CRTA}}$ as defined in Equation (Equation 16) are also shown, together with the scaled averaged three-phonon iteraction strengths $\tilde{P}$ defined in Equation (Equation 18).

	κ (W m^−1^ K^−1^)				$\kappa /\tau^{\mathrm{CTRA}}$ (W m^−1^ K^−1^ ps^−1^)	$\tau^{\mathrm{CRTA}}$ (ps)	$\tilde{P}\times {\left(3n_{a}\right)}^{2}$ (10^−8^ eV^2^)
	$\kappa_{\mathrm{xx}}$	$\kappa_{\mathrm{yy}}$	$\kappa_{\mathrm{zz}}$	$\kappa_{\mathrm{ave}}$			
d-Si	136.24	-	-	136.24	5.002	27.24	2.164
C-II	6.33	-	-	6.33	0.458	13.81	2.704
*o*C24	57.86	44.70	20.21	40.92	2.295	17.83	2.983
K-II/C-I	43.54	-	-	43.54	0.829	52.52	1.184
K-V/C-VI	38.24	37.68	32.96	36.29	0.815	44.53	1.327
K-VII/C-V	30.45	-	32.56	31.16	0.770	40.45	1.477

## Data Availability

Raw data from this study, including optimised structures, input and output files for the Phonopy and Phono3py codes, and sample input files for the Vienna Ab initio Simulation Package code, will be made available to download free of charge from an online repository at https://doi.org/10.17632/9hywbzt8zd, accessed on 20 September 2022.

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
