# Peer review of "Structural Dynamics, Phonon Spectra and Thermal Transport in the Silicon Clathrates"

_molecules, 2022, doi:10.3390/molecules27196431_

Round 1

Reviewer 1 Report

Achieving the goal to reduce greenhouse gas emissions requires technologies for effective clean energy sources. As most part of the energy used is wasted as heat research interest in thermoelectric power is growing. The requirement to optimize thermoelectric properties led to toxic materials containing heavy metals. Results of research to identify more abundant and less toxic candidate thermoelectrics with good electrical transport characteristic of crystalline semiconductors and poor thermal transport characteristic of glasses is encapsulated in the PGEC (Phonon-Glass-Electron-Crystal) concept.

An important subclass of potential PGEC compounds are the inorganic clathrates, i.e. open-structured 3D frameworks composed of polyhedral cages, and the wide variety of possible structural motifs lead to a complex relationship between the cage structure, structural dynamics, and the thermal conductivity. Ab initio calculations are playing an important role in understanding of the origin of high-performance thermoelectrics.

In the paper, periodic DFT (Density-Functional-Theory) calculations are utilized to investigate the structural dynamics of bulk diamond silicon and 5 framework Si-structures. There are modeled and analyzed phonon and reference IR (InfraRed) and Raman spectra; compared the Si allotropes energetic stabilities; calculated the lattice thermal conductivities of these 6 materials. Such an improved understanding of the factors that determine the thermal transport in clathrates and related systems would inform the future design of high-performance thermoelectrics.

In particular, authors predict a very low room-temperature lattice thermal conductivity for a Si clathrate-structure in a good agreement with available experimental data. The limited heat transport is attributed to low group velocities and short lifetimes, and suggests that further studies on this phase may be fruitful.

Author Response

We thank the reviewer for their careful reading of our manuscript and for their positive comments on our work.

Reviewer 2 Report

The authors have investigated the Structural Dynamics, Phonon Spectra and Thermal Transport in the Silicon Clathrates via DFT and MD simulations Although the topic is a matter of interest, however a revision is needed before the manuscript being published

1. The introduction about ZT and thermoelectrics can be updated with some recent literature (Scientific Reports 11 (1), 1-10 2021)

2. Lattice thermal conductivity and its electronic part must be elaborated further.

3. Comparison of transport properties with other thermoelectric can add value to this article e.g Heuslers, perovskites Molecules 27 (12), 3785 2022 1

Author Response

We thank the reviewer for taking the time to read and comment on our manuscript. Our responses to their three points are as follows:

1. We thank the reviewer for their suggestion. We have modified the revised Introduction to include references to some recent review papers covering the different classes of thermoelectric materials: "This has led to wide-ranging research to identify more abundant and less toxic candidate thermoelectrics, including other chalcogenides, oxides and oxyclalcogenides alongside skutterudite and clathrate framework structures, intermetallics and half-Heuslers, among others."

2. We are unsure what the reviewer means by "Lattice thermal conductivity and its electronic part must be elaborated further." Section 3.4 presents a detailed description of how the lattice thermal conductivity is calculated and discusses the heat transport in the clathrates in terms of contributions from phonon group velocities (harmonic effects) and lifetimes (anharmonic effects). In this study we focus on the structural dynamics (phonons) in the materials and we therefore do not examine the electronic structure nor calculate the electrical transport properties. We have however added the following statement to elaborate the link between the carrier concentration and electrical transport properties in typical thermoelectric materials: "In particular, increasing n increases the conductivity but often decreases the Seebeck coefficient and increases the electronic contribution to the thermal conductivity, which in heavily-doped semiconductors is often proportional to σ through the Wiedemann–Franz law."

3. As noted in #2 above, we have not investigated the full set of transport properties of these materials. While our study is motivated by the potential of Si clathrates as candidate thermoelectrics, our primary focus is on how the different framework structures impact the structural dynamics and lattice thermal conductivity, and we consider a detailed comparison with other TEs to be out of scope.

We would also like to note that the citation to the first of the two articles the reviewer recommend citing was insufficient to look it up (authors, article number and/or a DOI are needed). We believe we were able to find it through the reference list from the second article the reviewer recommends, by virtue of it having the same corresponding author: 10.1038/s41598-021-00314-6.

The second article the reviewer recommends does not appear to predict any thermoelectric transport properties but rather establishes the stability of SrBaSn under strain. It is difficult to see how this is relevant to our study. The first article predicts the properties, including thermal conductivity, of a number of half-Heuslers. As noted in #3 above, we believe comparing the predicted thermal conductivity of the clathrates examined in this work with other thermoelectric materials is out of scope.

Reviewer 3 Report

The manuscript is nicely written and the results will help researchers to understand the indepth concept of thermoelectricity. The manuscript can be accepted in its present form. 

Author Response

We thank the reviewer for their time and for their recommendation to publish without change.